# Analysis of circulating-microRNA expression in lactating Holstein cows under summer heat stress

Jihwan Lee[1], Soohyun Lee[2], Junkyu Son[1], Hyeonju Lim[1], Euntae Kim[1], Donghyun Kim[1], Seungmin Ha[1], Taiyoung Hur[1], Seunghwan Lee[2]*, Inchul Choi[2]*

**1** Dairy Science Division, National Institute of Animal Science, RDA, Cheon-an, Republic of Korea,
**2** Department of Animal and Dairy Sciences, Chungnam National University, Daejeon, Republic of Korea

* icchoi@cnu.ac.kr (IC); slee46@cnu.ac.kr (SL)

**Data Availability Statement:** All relevant data are within the manuscript and its Supporting Information files. Raw sequencing reads of circulating miRNAs are publicly available on the GEO database (accession number GSE150912).

## Abstract

Korean peninsula weather is rapidly becoming subtropical due to global warming. In summer 2018, South Korea experienced the highest temperatures since the meteorological observations recorded in 1907. Heat stress has a negative effect on Holstein cows, the most popular breed of dairy cattle in South Korea, which is susceptible to heat. To examine physiological changes in dairy cows under heat stress conditions, we analyzed the profiles circulating microRNAs isolated from whole blood samples collected under heat stress and non-heat stress conditions using small RNA sequencing. We compared the expression profiles in lactating cows under heat stress and non-heat stress conditions to understand the regulation of biological processes in heat-stressed cows. Moreover, we measured several heat stress indicators, such as rectal temperature, milk yield, and average daily gain. All these assessments showed that pregnant cows were more susceptible to heat stress than non-pregnant cows. In addition, we found the differential expression of 11 miRNAs (bta-miR-19a, bta-miR-19b, bta-miR-30a-5p, and several from the bta-miR-2284 family) in both pregnant and non-pregnant cows under heat stress conditions. In target gene prediction and gene set enrichment analysis, these miRNAs were found to be associated with the cyto-skeleton, cell junction, vasculogenesis, cell proliferation, ATP synthesis, oxidative stress, and immune responses involved in heat response. These miRNAs can be used as potential biomarkers for heat stress.

## Introduction

Following global warming, including the northward expansion of the subtropical climate zone, South Korea, located at the northern hemisphere of East Asia, is susceptible to the impact of climate change [1–3]. For example, the average increase in temperature in South Korea was 1.7°C, while global temperature increased by 0.7°C from 1912 to 2008 [2, 4]. Mainly, in summer 2018, South Korea experienced extreme hot temperatures since the meteorological observations recorded in 1907. Moreover, heat stress (HS) has a negative influence on livestock

**Funding:** This research was supported by a grant from National Institute of Animal Science(No. PJ014392), Republic of Korea. The funders had no role in study design, data collection and analysis, decision to publish, or preparation of the manuscript.

**Competing interests:** The authors have declared that no competing interests exist.

productivity, particularly, Holstein Friesian, the most popular breed of dairy cattle in South Korea. They are more sensitive to HS as it induces hormonal changes, infection, metabolic disorders, and abnormal embryo development in dairy cattle [5, 6], consequently affecting the economic traits such as growth, milk production, and infertility. HS can lead to limited feed-intake and imbalanced hormone secretion, resulting in a decrease in growth and reproductive efficiency. For example, the placental function of heat-stressed cows during late gestation is impaired by decreased secretion of placental hormones such as estrone sulfate, leading to retarded fetal growth and low birth weight of the calves [7]. Moreover, the reproductive performance of Holstein cow is more susceptible to HS during summer, suggesting that HS conditions, including elevated temperature and humidity, decrease thermal tolerance [8]. Jiangijing Liu et al. (2019) reported that the pregnancy rate of Holstein cows is 39.4% at temperature-humidity index (THI) < 72 (non-HS) and decreased to 31.6% at THI > 78.0 (Intermediate HS) [9]. In Florida, pregnancy rates of lactating cows in the summer is low (13.5%) [10]. Furthermore, the number of mounts per estrus also decreased by nearly half in summer, compared to winter [11]. For lactating cows, HS conditions have been reported to reduce milk yield by about 30–40% [12–14].

In addition to physiological responses, economic losses, including the cost of veterinary care and farm management (fans, sprinklers installation) and involuntary culling can have negative impacts on the dairy industry [15–17]. Therefore, the development of feasible methods and identification of biomarkers are essential for recognizing heat-stressed cows in order to provide individual attention and tailor-made care. To investigate the effects of heat stress on dairy cows, we used profiles of circulating microRNAs isolated from whole blood that was collected in HS and non-HS season. Recent studies have demonstrated that microRNAs (miR-NAs) are exported to the extracellular environment through microvesicles such as exosomes and circulate in the blood [18, 19] and have shown tremendous potential as non-invasive biomarkers in human cancers [20–22] and pregnancy [23], estrus [24], and aging [17] in dairy cattle. In addition, miRNAs have different expression patterns under environmental and physiological changes such as HS [25, 26]. The goal of this study was to find the potential biomarkers related to HS in lactating dairy cows and to identify the association of candidate miRNAs with putative targeted genes under HS.

## Materials and methods

### Experimental animals

Before the start of the experiments, veterinarians regularly checked the cows' medical condition in the dairy research center, and nine lactating Holstein-Friesian cows determined to be healthy and free of disease were selected. All cows had 227±45.5 (mean±standard deviation) average milking days (Individual cow records including age, parity, and calving date in S1 Table in S2 File). Diet was formulated according to NRC 2001, and the cows were fed twice a day to meet the nutrient requirement. Pregnant (n = 4) and non-pregnant (n = 5) cows ate the same feed in the same area. Freshwater was available for free, and mineral blocks were placed on columns of the barn. In this study, all animal experimental designs and procedures were approved by the National Institute of Animal Science Animal Care and Ethics Committee in South Korea (NIAS-109). After the experiment, cows were housed for further studies.

### Heat stress indicators

**Temperature-humidity index (THI).** In order to measure the THI in the barn, a THI measuring device (Testo-174d, 5720500, Germany), which automatically recorded the temperature and relative humidity, was attached to columns in three points in the barn. The

measurements were recorded a total of 12 times a day at 2 h intervals. THI was calculated using the following formula [27, 28]: THI = (0.8 × temperature (˚C) + [(relative humidity (%)/100) × (temperature (˚C) − 14.4)] + 46.4.

**Body weight & milk yield measurement.** Bodyweight and milk yield were recorded automatically during the milking by the milking robot (Lely Astronaut milking robot, Netherlands) from June to October. We analyzed the average daily gain (ADG) using bodyweight data. Based on the average milk yield for May, the relative average milk yield for each cow from June to October was calculated.

**Rectal temperature.** Rectal temperature was measured at the same time as the blood collection. In order to measure rectal temperature, the feces of cows were removed. Using rectal thermometer (POLYGREEN Co. Ltd, Germany), the rectal temperature was manually measured at 14:00 under HS (Heat Stress; THI: 86.29) and NHS conditions (Non-Heat Stress; THI: 60.87). For the accuracy, the measurement was repeated three times per cow, and the rectal thermometer was inserted into the rectum more than 15 cm deep. We also calculated the heat tolerance coefficient (HTC) according to the method described by Road A.O [29]; HTC = 100-10(RT(˚F)-101).

**Statistical analysis.** The quantitative data are presented as mean± standard error of the mean (s.e.m) and analyzed by using GraphPad Prism (ver. 5.03, GraphPad Software, San Diego, CA, USA). Significant differences were determined by Student t-test or one-way ANOVA. P values <0.05 were considered statistically significant differences unless otherwise stated.

## Blood collection

The all cows were kept inside the barn that was opened for natural ventilation. We calculated daily THI using the recorded air temperature and humidity, and chose the sampling date when daily minimum THI > 72 and daily maximum THI <72 lasted for more than four weeks (Fig 1). Whole blood was collected separately from jugular vein of the same cows (n = 9) at two different environmental seasons (summer and autumn) using PAXgene Blood RNA tube (2.5 ml/cow; Qiagen, 762165, California, USA) and vacutainer tube containing sodium heparin (10 ml/cow; BD, vacutainer®, 367874, Franklin Lakes, NJ, USA). PAXgene Blood RNA tubes were stored at -80˚C until miRNA extraction. Heparin tubes were immediately centrifuged at 3,000×g for 10 min at 4˚C.

## MiRNA extraction and cDNA synthesis

MiRNAs were isolated from whole blood using the PAXgene Blood MicroRNA Kit (Qiagen) according to the manufacturer's instructions. The miRNA concentrations were determined by using NanoDrop (Optigen NANO Q, South Korea), and cDNA was synthesized using the miScript II RT Kit (Qiagen, 218160, California, USA) following the manufacturer's instructions and stored at -80˚C until use. Realtime-qPCR was performed on 11 differentially expressed (DE) miRNAs based on miRNA-seq results (|FC| ≥ 2, P < 0.05) using miScript SYBR Green PCR Kit (Qiagen 218073, California, USA) according to the manufacturer's instructions with StepOne Applied Biosystems real-time PCR machine (Applied Biosystems, Foster City, CA). All RT-qPCR reactions were performed in triplicates. Endogenous control was bta-miR-128 [23]. All primer information used in this experiment is represented in S2 Table in S2 File.

## miRNA-sequencing experiment and statistical analysis

We checked miRNA integrity using an Agilent 2100 Bioanalyzer (Agilent Technologies, Santa Clara, CA, USA) with an RNA integrity number greater than or equal to 7. To construct the

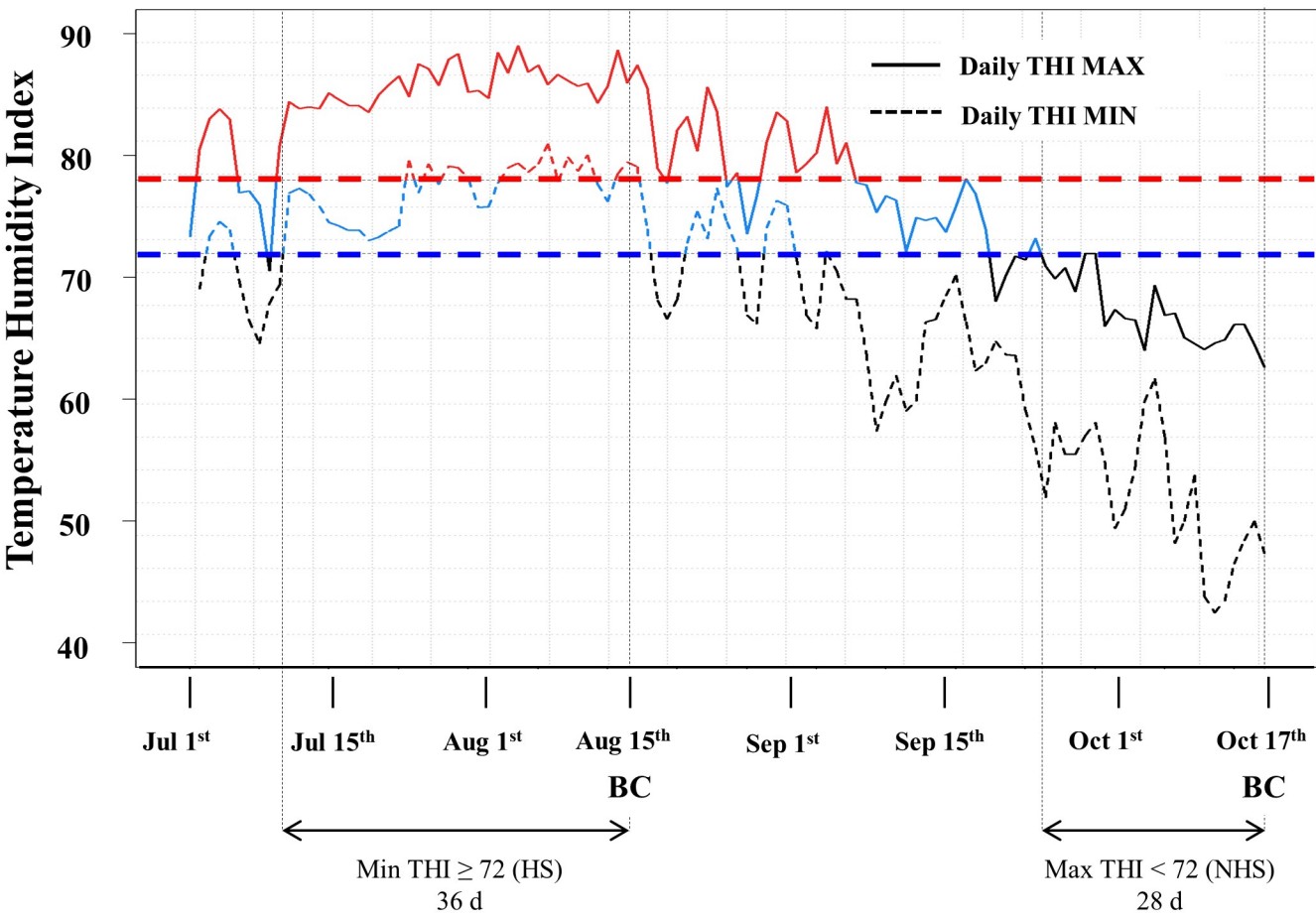

**Fig 1. Temperature-humidity index (THI) measured on dairy barn.** The straight line represents daily THI maximum and the dotted line represents daily THI minimum. THI over 78 is marked in red (moderate to severe stress), and 72–78 in blue (mild to moderate stress), and less than 72 in black (non-stress). Vertical dotted line represents THI on the day of sampling. BC, Blood collection; HS, Heat stress condition; NHS, Non-heat stress condition.

library, we performed adapter ligation, reverse transcription, PCR amplification and pooled gel purification using Truseq Small RNA Library Prep Kit (Illumina, San Diego, USA). A flow cell containing millions of unique clusters was added to Illumina Hiseq2000 sequencer. Raw sequencing reads of circulating miRNAs (publicly available on the GEO database accession number GSE150912) extracted from all samples were pre-processed and analyzed using miR-Deep2 software. Adapter trimming was performed to remove the adapter sequences attached to the miRNA during small RNA library construction process using Cutadapt v.1.9.1 and then to increase accuracy, trimmed reads (minimum 18 bp) were collected to form a cluster. The pre-processed and clustered reads were aligned with Mus musculus reference genome, and then those reads were aligned with Mus musculus precursor and matured miRNAs extracted from miRBase v21. To detect known and novel miRNAs and estimate their abundance, we used miRDeep2 software. The differentially expressed gene analysis was performed by Limma-voom v3.34.9 R package [30–32]. Before the analysis, genes which had raw read counts were filtered out by 'filterByExpr' function in edgeR R package [33]. Furthermore, TMM normalization was performed to normalize each library size using 'calcNormFactors' function in edgeR. Next, we used 'voom' function of Limma for read counts to transform to logarithmic (base 2) scale prior to linear modeling. Finally, empirical Bayes and moderated t-test were used to

detect differentially expressed genes between two groups. The threshold to identify differentially expressed genes was set up to the p-value < 0.01 and the logarithm fold change |logFC| > 2.

## Bioinformatics analysis

Two databases, miRmap (v1.1, mirmap.ezlab.org) and TargetScan (v7.2, targetscan.org), were used to predict the target genes for DE miRNAs [34, 35]. Target genes were selected based on the miRmap Score ≥ 80 (provided by miRmap program) and context++ score percentile ≥ 95 (provided by TargetScan program) was taken as the cut-off value to increase the accuracy of the analysis. Gene Ontology analysis was performed using the PANTHER Classification System (v.14.1) [36] to identify the functional enrichment for a gene set. In addition, we analyzed the Kyoto Encyclopedia of Genes and Genomes (KEGG) pathways by uploading the target gene list to DAVID Bioinformatics Resources 6.8 [37] to identify the functional enrichment signaling pathways related to these target genes.

# Results

## Estimation of THI and blood collection

We measured ambient temperature and relative humidity inside the barn daily to estimate THI (Fig 1). We observed that minimum THI exceeded 72 from the first week of July and even reached over 80, and moderate/severe HS conditions (THI > 78) lasted for over one month. Both maximum and minimum THI peaked around mid-August, and gradually declined; however, mild to moderate HS (THI 72–78) was detected until the end of September. We collected whole blood at 14:00 from both pregnant and non-pregnant cows in the summer (HS) and autumn (NHS). The THI ranged from 79.10 to 87.73 (14:00; 86.29) and 47.3–64.85 (14:00; 60.87) at the HS and NHS sample collecting day. The minimum THI of more than 72 (the cut-off level for HS) until the HS sampling lasted for 36 days and the maximum THI of less than 72 until NHS sampling lasted for 28 days.

## Effects of HS on physiological changes

We measured physiological HS indicators such as rectal temperature, milk yield, and ADG of weight in pregnant and non-pregnant cows. The rectal temperature of cows was measured at the time of blood collection. In NHS conditions, there were no differences between pregnant (38.4˚C±0.07) and non-pregnant cows (38.02˚C±0.14). However, the rectal temperatures of pregnant cows (40.15˚C±0.17) were higher than that of non-pregnant cows (39.36˚C±0.1, P < 0.05, Table 1) under HS conditions. We also observed similar results in the HTC test; non-pregnant cow showed higher tolerance in both HS and NHS conditions, and higher tolerance in non-pregnant under HS conditions were observed, compared to NHS conditions (Table 1).

**Table 1. Rectal temperature and HTC values under environmental conditions on the day of sampling.**

| Group | | Rectal temp.(˚C) | HTC | Temperature Range (˚C) | THI |
|---|---|---|---|---|---|
| **HS** | Pregnancy | 40.15±0.17[a] | 67.3±0.3[a] | 27.3–37.4 | 79.10–87.73 |
| | Non-pregnancy | 39.36±0.11[b] | 81.52±0.20[b] | | |
| **NHS** | Pregnancy | 38.40±0.07[c] | 98.8.±0.12[c] | 8.4–17.6 | 47.3–64.85 |
| | Non-pregnancy | 38.02±0.14[c] | 105.64.±0.25[d] | | |

HS, Heat stress; NHS, Non-heat stress; HTC, Heat tolerance coefficient: 100–10×(RT(˚F)-101).

To estimate the effect of HS on milk production, we used relative average milk yield; the ratio of the total milk yield in a month of the testing period to the yield of a month (May). Relative average milk yield (AMY) decreased gradually in both pregnant and non-pregnant cows, compared to milk yield of May, particularly a significant reduction was observed in pregnant cows during extremely hot summer (August), and the recovery of AMY was observed in September (Fig 2A), indicating HS more severely may affect milk production of pregnant cows. To investigate the effects of HS on growth performance, we analyzed the changes in ADG of pregnant and non-pregnant cows during testing periods. As shown in Fig 2B, we found an increase in ADG in both pregnant and non-pregnant cows, except for HS pregnant cows during July, although ADG during summer was significantly reduced.

## Identification of DE miRNAs of HS cows

We analyzed miRNAs isolated from the whole blood to identify DE miRNAs between heat-stressed and non-heat-stressed cows using small RNA sequencing. In the non-pregnant group, we found that 23 miRNAs were significantly DE ($\geq$ 2-FC in the expression compared to NHS controls; P < 0.05), including nine, upregulated and 14 downregulated miRNAs (Table 2). In the pregnant group, we detected 28 DE miRNAs (10 upregulated, 18 downregulated; $\geq$ 2-FC and P < 0.05, Fig 3). We identified 11 common DE miRNAs in both non-pregnant and pregnant cows; two miRNAs (bta-miR-19a and bta-miR-19b) were upregulated and nine, including bta-miR-30a-5p and several bta-miR-2284 families, were downregulated (Table 3). We

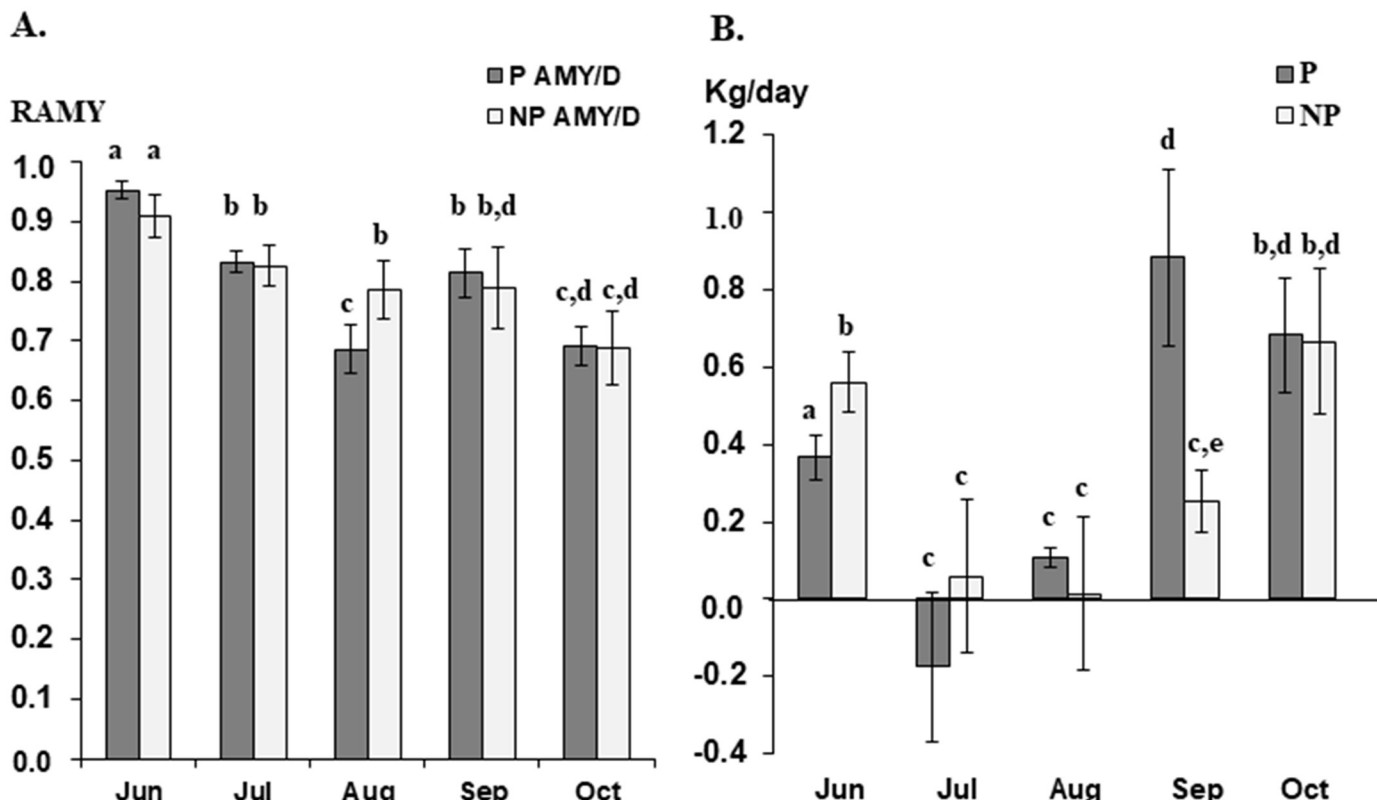

**Fig 2. Physiological heat stress indicators recorded during HS and NHS conditions in both pregnant and non-pregnant lactating cows.** (A) Relative average milk yield (RAMY) to May; (B) Average dairy gain (ADG). HS, Heat stress; NHS, Non-heat stress; P, Pregnancy; NP, Non-pregnancy. Different superscript letters indicate significant difference (P<0.05).

**Table 2. Differentially expressed miRNAs in pregnant and non-pregnant groups under heat-stressed condition.**

| Status | Mature_miRNA | RNA sequencing |
|---|---|---|
| | | Fold change \|FC\| (HS/NHS) ≥ 2 (P<0.05) |
| **NP** | | FC value: Up-regulation |
| | Bta-miR-19a | 3.73 |
| | Bta-miR-19b | 3.01 |
| | Bta-miR-153 | 2.64 |
| | Bta-miR-301a | 2.32 |
| | Bta-miR-370 | 5.63 |
| | Bta-miR-374a | 2.81 |
| | Bta-miR-454 | 2.01 |
| | Bta-miR-2285h | 2.26 |
| | Bta-miR-6119-5p | 2.18 |
| | | FC value: Downregulation |
| | Bta-miR-30a-5p | -2.77 |
| | Bta-miR-133a | -10.17 |
| | Bta-miR-151-3p | -2.10 |
| | Bta-miR-2284a | -11.04 |
| | Bta-miR-2284b | -2.87 |
| | Bta-miR-2284h-5p | -4.56 |
| | Bta-miR-2284k | -5.47 |
| | Bta-miR-2284v | -2.16 |
| | Bta-miR-2284w | -2.20 |
| | Bta-miR-2284x | -4.87 |
| | Bta-miR-2284y | -4.71 |
| | Bta-miR-2332 | -2.33 |
| | Bta-miR-2424 | -3.17 |
| | Bta-miR-2453 | -4.13 |
| **P** | | FC value: Up-regulation |
| | Bta-miR-19a | 3.49 |
| | Bta-miR-19b | 3.34 |
| | Bta-miR-20b | 2.40 |
| | Bta-miR-29d-3p | 3.77 |
| | Bta-miR-106a | 2.25 |
| | Bta-miR-378d | 2.39 |
| | Bta-miR-497 | 4.24 |
| | Bta-miR-502a | 2.46 |
| | Bta-miR-2285ad | 2.44 |
| | Bta-miR-2285o | 5.98 |
| | | FC value: Downregulation |
| | Bta-miR-30a-5p | -4.44 |
| | Bta-miR-146b | -2.08 |
| | Bta-miR-296-3p | -2.11 |
| | Bta-miR-1246 | -9.49 |
| | Bta-miR-2284a | -17.52 |
| | Bta-miR-2284aa | -2.31 |
| | Bta-miR-2284ab | -2.40 |
| | Bta-miR-2284b | -3.61 |

(*Continued*)

**Table 2.** (Continued)

| Status | Mature_miRNA | RNA sequencing |
|---|---|---|
| | | Fold change |FC| (HS/NHS) ≥ 2 (P<0.05) |
| | Bta-miR-2284h-5p | -8.05 |
| | Bta-miR-2284k | -7.77 |
| | Bta-miR-2284r | -2.82 |
| | Bta-miR-2284v | -2.84 |
| | Bta-miR-2284w | -2.69 |
| | Bta-miR-2284x | -10.37 |
| | Bta-miR-2284y | -9.95 |
| | Bta-miR-2284z | -2.19 |
| | Bta-miR-2397-5p | -2.08 |
| | Bta-miR-2457 | -2.15 |

also validated these DE miRNAs by qRT-PCR. The results were very similar to the small RNA sequencing results except for bta-miR-2284x (Table 3).

## Putative target gene and signaling pathway analysis

We analyzed putative target genes of 11 common DE miRNAs using miRmap and TargetScan, identified 890 genes, and performed gene set enrichment analysis (GSEA) using DAVID and

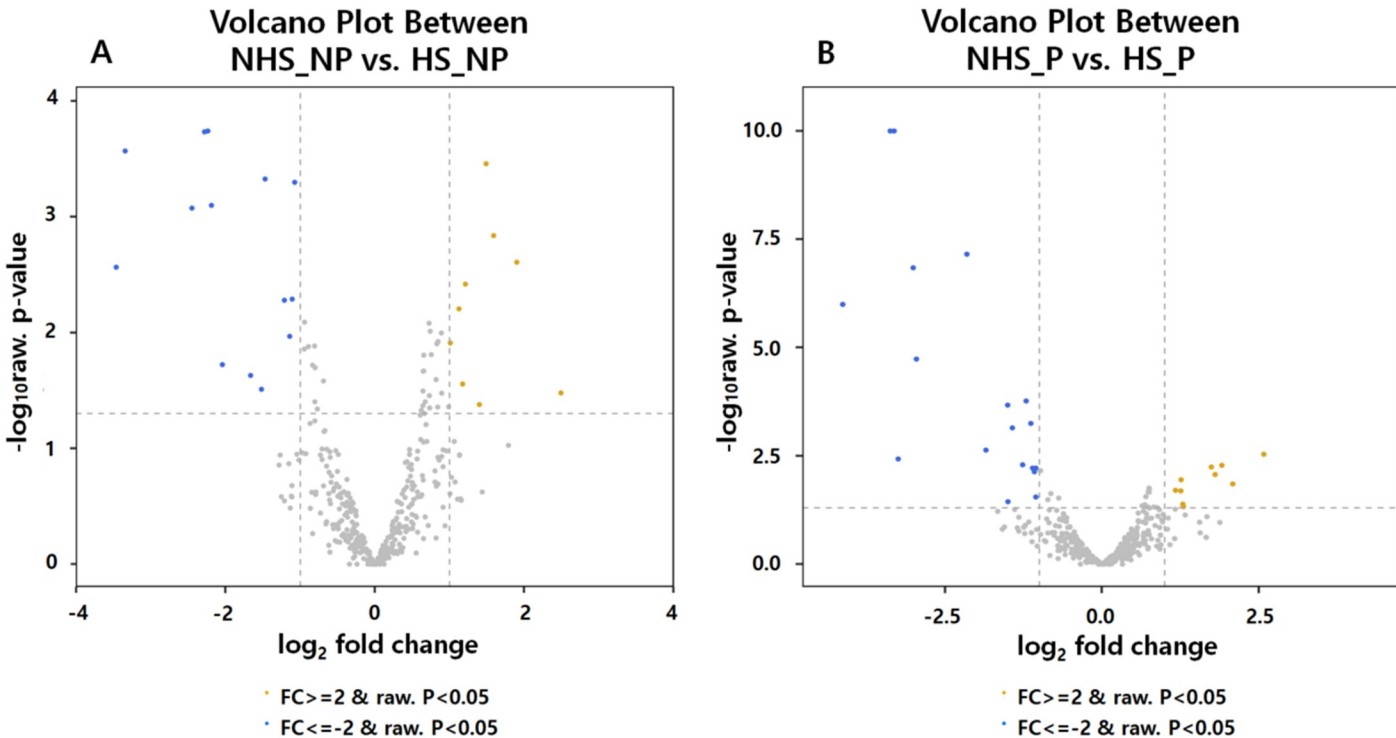

**Fig 3. Volcano plot showing differentially expressed miRNAs between NHS and HS using transformed normalized data.** |Fold change| value ≥ 2 and P < 0.05 are represented different colors (minus: blue, plus: yellow). (A) Differentially expressed miRNA values in non-pregnant cows; (B) Differentially expressed miRNA values in pregnant cows; HS, Heat stress; NHS, Non-heat stress; P, Pregnancy; NP, Non-pregnancy.

**Table 3. Differentially expressed miRNAs in both non-pregnant and pregnant cows under HS conditions (P < 0.05).**

| Mature_miRNA | RNA sequencing | RT-qPCR |
|---|---|---|
| | Fold change \|FC\| (HS/NHS) ≥ 2 (P<0.05) | Endogenous Control: bta-miR-128 |
| bta-miR-19a | 3.61 | 2.01 |
| bta-miR-19b | 3.17 | 2.14 |
| bta-miR-30a-5p | -3.61 | 0.75 |
| bta-miR-2284a | -14.28 | 0.42 |
| bta-miR-2284b | -3.24 | 0.27 |
| bta-miR-2284h-5p | -6.30 | 0.84 |
| bta-miR-2284k | -6.62 | 0.98 |
| bta-miR-2284v | -2.50 | 0.35 |
| bta-miR-2284w | -2.45 | 0.63 |
| bta-miR-2284x | -7.62 | 1.08 |
| bta-miR-2284y | -7.33 | 0.79 |

HS, heat stress; NHS, non-heat stress.

PANTHER (S1 Fig in S1 File). The GSEA showed that the putative target genes were associated with the cytoskeleton, cell junction, immune response, oxidative stress involved in heat response (Table 4). Besides, the KEGG pathway showed 18 statistically significant pathways (S1-S3 Tables in S2 File). For example, the FoxO signaling pathway, peroxisome, regulation of actin cytoskeleton, TNF signaling pathway, Rap1 signaling pathway, and chemokine signaling pathway were found to be closely related to HS response. Furthermore, we analyzed putative target genes of pregnancy-specific 23 DE miRNAs (S2 Fig in S1 File). Several DE miRNAs (bta-miR-146b, bta-miR-20b, bta-miR-29d-3p, bta-miR-1246) were found to play essential

**Table 4. Predicted target genes related to heat stress responses of miRNAs differentially expressed both pregnant and non-pregnant cows.**

| DE miRNA | Related to heat stress | Target gene |
|---|---|---|
| **Bta-miR-19a, -19b** | Cytoskeleton | CLIP1, RAP2C, S1PR1, DNAI1, LPP, MICAL2, WASF3 |
| | Cell junction | GJA1 |
| | Vasculogenesis | ETV5, ANGPTL1, PTGER2 |
| | Cell proliferation | VPS37A, POSTN, MAPK14, MAP3K12 |
| | Immune response | TNF, MAPK14, MAP3K12, ALOX12, C5, BCL3 |
| | Oxidative and heat stress response | HSPBAP1, UCP3 |
| | ATP synthesis | COQ10B |
| | DNA repair | IGFBP3 |
| **Bta-miR-30a-5p** | Cytoskeleton | RAP2C |
| | Vasculogenesis | ANGPTL1 |
| | Cell proliferation | MAP3K12, KDHRBS3 |
| | Immune response | FAP, ITK, AHSA2, PNKD |
| | Oxidative stress | UCP3, PLA2R1 |
| | Reproduction | ADAM19 |
| **Bta-miR-2284 family** | Numerous genes involved in the immune responses (Data are not shown) | |

DE miRNAs, differentially expressed miRNAs.

**Table 5. Predicted target genes related to heat stress responses of miRNAs differentially expressed in pregnant cows.**

| DE miRNA | Target genes | Responses |
|---|---|---|
| Bta-miR-146b | CCL11 | Improvement of Corpus luteum function |
| Bta-miR-20b | XCL1 | |
| Bta-miR-29d-3p | COL2A1, COL4A1, COL4A5, COL6A3, COL11A1 | |
| Bta-miR-1246 | StAR | Progesterone biosynthesis |

DE miRNAs, differentially expressed miRNAs.

roles in the regulation of progesterone biosynthesis and corpus luteum (Table 5). KEGG pathway showed statistically significant 24 pathways (S4 Table in S2 File). Interestingly, the prolactin signaling pathway was found to be closely related to progesterone synthesis.

## Discussion

THI is widely used as an indicator to estimate the degree of HS in livestock animals because the valid and reliable assessment of heat load may mitigate or minimize economic loss such as inefficient reproductive performance and milk production in dairy cattle [38, 39]. The THI values can be catabolized into five different classes; no HS (THI < 72), mild HS ($72 \leq$ THI $\leq$ 78), moderate HS (78 < THI < 89), severe HS ($89 \leq$ THI $\leq$ 98), and death (THI > 98) [40, 41]. To confirm whether cows experienced HS, we checked the association between THI and physiological changes such as milk yield and ADG. We collected whole blood when mild HS lasted more than one month (precisely 36 days), where milk yield and ADG decreased, and non-HS samples were obtained at four weeks after minimum THI returned to < 72 (non-HS condition), implying heat-stressed cows may fully recover from the long-term HS as seen in increased milk production and ADG.

Interestingly, a significant decrease in milk production of pregnant cows during summer might be attributed, in part, to feed-intake reduction because the lactating and pregnant cows need more energy not only for the milk production but also for fetal growth, compared to non-pregnant lactating cows [42, 43]. In this study, we indirectly estimated the feed-intake using ADG. Notably, the negative ADG observed in pregnant HS cows suggested that HS may cause appetite suppression and/or lower feed-intake, consequently resulting in the loss of body weight. The dramatic increase in ADG may be attributed to compensatory placental growth by increased feed-intake.

In accordance with previous studies, where a positive correlation between the rectal temperature and THI value was reported [27], we also observed higher rectal temperature and lower HTC in pregnant HS cows, indicating pregnant cows were more sensitive to HS than non-pregnant cows. Physiological indicators proved that pregnant cows were more susceptible to thermal stress, compared to non-pregnant cows.

To further understand the association between these physiological indicators, biological processes, and cellular responses to HS, we identified DE miRNAs using RNAseq. We employed an *in silico* approach for miRNA target prediction because circulating miRNAs in body fluid may play essential roles in all biological processes and present as potentially useful biomarkers of the HS response. We analyzed 11 common DE miRNAs in both pregnant and non-pregnant cows under HS conditions (Table 3). Two upregulated bta-miR-19a and 19b have been previously reported to target *HSPBAP1*, *DNAJB1*, and *HPX* that respond to heat stress, and downregulated bta-miR-30a-5p may have potential roles in heat stress response,

such as oxidation, through its target genes, *PLA2R1* and *PICEN* [25]. In target gene prediction and GESA, bta-miR-19a and bta-miR-19b are associated with biological functions including the cytoskeleton, cell junction, vasculogenesis, cell proliferation, oxidative stress, immune response and ATP synthesis (Table 4). It is well-established that mammalian cells, such as the oocyte, embryo, and mammary gland epithelium were subjected to heat shock. Mainly, degradation and dysfunction of the cytoskeleton in response to HS may result in the aberrant mitochondrial distribution, impaired mitochondrial function, and apoptosis/necrosis. Cellular junctions are also involved in the transportation of ions and small molecules between the blood and milk barrier at the mammary gland. Interestingly, we identified that bta-miR-30a-5p regulated a disintegrin and metalloprotease 19 (*ADAM19*), which is required for early embryo development and implantation in mammals [44]. A ruminant specific miRNA, bta-miR-2284 family, was also identified, putative target genes of which are related to the heat-induced immune response, although the biological roles have not been determined [45].

Integrative approaches using the predicted target genes of DE miRNAs, and DAVID identified several KEGG pathways such as FoxO signaling, actin cytoskeleton, Rap1 and rapamycin (mTOR) signaling that are involved in transcription activation of heat shock proteins, mitochondria distribution and activity, regulation of energy homeostasis including glucose, and lipid metabolism [46–54]. We also found, in line with a previous study [25], immune response related pathways such as chemokine and TNF signaling [55, 56]. In addition, several DE miRNAs (bta-miR-146b, bta-miR-20b, bta-miR-29d-3p, bta-miR-1246) were differentially expressed in summer heat-stressed pregnant cows. Interestingly, bioinformatics analysis showed their predicted target genes are associated with function of corpus luteum including progesterone biosynthesis [57, 58] and prolactin signaling pathways (S4 Table in S2 File).

As described above, the blood samples were collected from the same animals including pregnant and non-pregnant cows, suggesting that the different period or gestational age may affect the profile of gene expression. Thus, we first identified differential expressed genes in both pregnant and non-pregnant cows under HS, reflecting effects of HS and different time periods including different stage of lactation. However, the effects may be canceled out by the common genes both pregnant and non-pregnant cows, and their parities with different stages of lactation (S1 Table in S2 File). In addition, profiles of putative target genes by DE miRNAs are also associated with metabolism and reproductive system, supporting that HS may induced the DE miRNAs. Comprehensively, our findings suggest that the experimentally verified miRNA targets and *in silico* analysis reflect that the selected miRNA could be potentially used to determine response to HS.

## Conclusions

We analyzed physiological HS indicators such as milk yield, rectal temperature, ADG, in pregnant and non-pregnant cows and found that pregnant cows are more vulnerable under HS conditions. In transcriptome analysis of miRNAs using RNA-sequencing, 11 miRNAs (bta-miR-19a, bta-miR-19b, bta-miR-30a-5p, several from bta-miR-2284 family) were differentially expressed ($|FC| \geq 2$, $P < 0.05$) in both pregnant and non-pregnant cows. In pregnant cows under HS, two miRNAs (bta-miR-20b, bta-miR-29d-3p) were upregulated while two miRNAs (Bta-miR-146b, bta-miR-1246) were downregulated. These selected miRNAs could be potential biomarkers associated with HS. Admittedly, the our findings of the present study are limited by the small datasets, their statistical approach, experimental designs that could be attributed to the inherent limitations of the samples (synchronized gestation ages, parities, lactation stages) and heat stress condition (frequent, intense, and duration). Thus, we need to verify our results using model animals or the climate control system (temperature and humidity).

## Supporting information

**S1 File.**
(EGG)

**S2 File.**
(EGG)

## Acknowledgments

We thank the Dairy Science Division at National Institute of Animal Science for helping us work on this project and JH Kim, a researcher at Macrogen, NGS company located in Daejeon, South Korea for helping us with miRNA-sequencing analysis.

## Author Contributions

**Conceptualization:** Jihwan Lee, Inchul Choi.

**Data curation:** Jihwan Lee, Inchul Choi.

**Formal analysis:** Soohyun Lee, Seunghwan Lee.

**Funding acquisition:** Jihwan Lee.

**Investigation:** Jihwan Lee.

**Methodology:** Jihwan Lee, Inchul Choi.

**Project administration:** Jihwan Lee.

**Resources:** Jihwan Lee, Junkyu Son, Hyeonju Lim, Euntae Kim, Donghyun Kim, Seungmin Ha, Taiyoung Hur, Inchul Choi.

**Software:** Jihwan Lee, Soohyun Lee, Seunghwan Lee, Inchul Choi.

**Supervision:** Inchul Choi.

**Validation:** Jihwan Lee, Inchul Choi.

**Visualization:** Jihwan Lee, Soohyun Lee, Seunghwan Lee, Inchul Choi.

**Writing – original draft:** Jihwan Lee, Inchul Choi.

**Writing – review & editing:** Jihwan Lee, Inchul Choi.

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
