## [Decision Letter · Decision Letter 0]

30 Apr 2020

PONE-D-20-07553

Analysis of circulating-microRNA expression in lactating Holstein cows under summer heat stress

PLOS ONE

Dear Assistant Professor Choi,

Thank you for submitting your manuscript to PLOS ONE. After careful consideration, we feel that it has merit but does not fully meet PLOS ONE’s publication criteria as it currently stands. Therefore, we invite you to submit a revised version of the manuscript that addresses the points raised during the review process.

THERE ARE NUMBER OF MAJOR ISSUES THAT NEED TO BE FIXED. IT IS IMPORTANT THAT A PROPER STATISTICAL ANALYSIS IS PERFORMED WITH THE TRANSCRIPTOME DATA. IT MUST INCLUDE MULTIPLE TESTING CORRECTION TO GUARD AGAINST TYPE 2 ERRORS.

We would appreciate receiving your revised manuscript by Jun 14 2020 11:59PM. To enhance the reproducibility of your results, we recommend that if applicable you deposit your laboratory protocols in protocols.io, where a protocol can be assigned its own identifier (DOI) such that it can be cited independently in the future. For instructions see: http://journals.plos.org/plosone/s/submission-guidelines#loc-laboratory-protocols

We look forward to receiving your revised manuscript.

Kind regards,

Juan J Loor

Academic Editor

PLOS ONE

2. In your Methods section, please provide additional details regarding the animals used in your study and ensure you have described the source.

3. In your Methods section, please state the volume of the blood samples collected for use in your study.

4. In your Methods section, please include a comment about the state of the animals following this research (i.e. housed for use in further research, etc.)

5. We note that you are reporting an analysis of a microarray, next-generation sequencing, or deep sequencing data set. PLOS requires that authors comply with field-specific standards for preparation, recording, and deposition of data in repositories appropriate to their field. Please upload these data to a stable, public repository (such as ArrayExpress, Gene Expression Omnibus (GEO), DNA Data Bank of Japan (DDBJ), NCBI GenBank, NCBI Sequence Read Archive, or EMBL Nucleotide Sequence Database (ENA)). In your revised cover letter, please provide the relevant accession numbers that may be used to access these data. For a full list of recommended repositories, see http://journals.plos.org/plosone/s/data-availability#loc-omics or http://journals.plos.org/plosone/s/data-availability#loc-sequencing.

Reviewers' comments:

Reviewer's Responses to Questions

**Comments to the Author**

1. Is the manuscript technically sound, and do the data support the conclusions?

Reviewer #1: Yes

Reviewer #2: No

2. Has the statistical analysis been performed appropriately and rigorously? 

Reviewer #1: Yes

Reviewer #2: No

3. Have the authors made all data underlying the findings in their manuscript fully available?

Reviewer #1: Yes

Reviewer #2: No

4. Is the manuscript presented in an intelligible fashion and written in standard English?

Reviewer #1: Yes

Reviewer #2: Yes

5. Review Comments to the Author

Reviewer #1: The manuscript entitled “Analysis of circulating-microRNA expression in lactating Holstein cows under summer heat stress” investigates the changes in circulatory (blood) microRNAs in response to heat stress in pregnant and non-pregnant lactating dairy cows. Although it is an attractive subject, some methodological issues need to be clarified.

Please explain in more detail: how many blood samples have been taken from the cows? Was there only one measurement per cow in spring and one in summer? Please specify when measurements were taken- what month and day were chosen?

As parameter of exposure to heat was chosen THI in barn. Were cows at any time exposed to the outside temperatures?

Pregnancy as a matter of investigation: were all pregnant cows of the same gestational age?

In the present model of investigation where comparison of blood samples of the same pregnant cows but in different time period (and different gestational age) were made in order to investigate factor-heat stress could be disputable. Namely, differences observed in different time periods could be related to pregnancy state and not heat stress. Differences in progesterone levels observed also, could be related to the stage of pregnancy and not necessary induced by heat stress.

Line 105: the authors stated: “Except for two cows, all cows were similar…” Were these differences taken into account considering results analysis?

Results: Statistical significances were not labeled neither in Table 1 nor in Figures

Line 224: it is not stated under what conditions significance was observed

Line 237: please rephrase, it is not clear at what time point significant difference was observed and please explain …”and milk increased after HS”? compared to what? That’s not what Figure shows.

Reviewer #2: Comments to Authors:

Major points

The authors do not seem to statistically analyze the large data set of RNA-seq appropriately, as shown in Line 185-186. For the large data set analysis, the authors should do multiple comparison and test the significance of the difference between the treatments by FDR, Bonferroni, or Benjamini criteria to avoid detection of false positives. They just describe ‘P < 0.05’, but do not mention the statistical analysis. This is of critical importance, which further affects not only subsequent screening and PCR of DE miRNAs, but also bioinformatics analyses. Using the incorrectly processed data, they will fail to reach correct appropriate conclusion and interpretation of the data, which will mislead the readers of this manuscript. Furthermore, the R-package for methods to visualize DE miRNAs is not provided.

In addition, comparison of the treatments is not described, for example, no animal numbers for each treatment.

Minor points

Line 41: Define miRNA.

Line 44: It is obvious that circulating miRNAs do not directly regulate gene expression in the cells of the target tissues. How do the authors explain the action of circulating miRNAs on these target cells?

Line 67: Holstein Friesian ?

Line 91: Define miRNA.

Line 104-105: the sentence ‘four of them were pregnant …..3.25.’ does not make sense.

Line 141: Define (spell out) NHS.

Line 163: Show the reason why the authors chose miR-128 for endogenous control.

6. PLOS authors have the option to publish the peer review history of their article (what does this mean?). If published, this will include your full peer review and any attached files.

Reviewer #1: No

Reviewer #2: No

---

## [Author Response · Author response to Decision Letter 0]

16 Jun 2020

Response to general comments

Re: According to the PLOS ONE’s style, we checked and rewrote. 

2. In your Methods section, please provide additional details regarding the animals used in your study and ensure you have described the source.

 Re: we provided additional information. Pls see the responses to reviewer 1 comments

3. In your Methods section, please state the volume of the blood samples collected for use in your study.

 Re: we described. Pls see the responses to reviewer 1 comments

4. In your Methods section, please include a comment about the state of the animals following this research (i.e. housed for use in further research, etc.)

 Re: After the experiment, cows were housed for further studies.

5. Accession number for RNAseq data:

Re: We added in method section

“ Raw sequencing reads of circulating miRNAs (publicly available on the GEO database accession number GSE150912)…” https://www.ncbi.nlm.nih.gov/geo/query/acc.cgi?acc=GSE150912

Reviewer #1: 

C1: Please explain in more detail: how many blood samples have been taken from the cows? Was there only one measurement per cow in spring and one in summer? Please specify when measurements were taken- what month and day were chosen?

Re: we added in more detail as follow (pls see underlines)

“The all cows were kept inside the barn that was opened for natural ventilation. We calculated daily THI using the recorded air temperature and humidity, and chose the sampling date when daily minimum THI > 72 and daily maximum THI <72 lasted for more than four weeks (Fig 1). Whole blood was collected separately from jugular vein of the same cows (n=9) at two different environmental seasons (summer and autumn) using PAXgene Blood RNA tube (2.5 ml/cow; Qiagen, 762165, California, USA) and vacutainer tube containing sodium heparin (10 ml/cow; BD, vacutainer®, 367874, Franklin Lakes, NJ, USA). PAXgene Blood RNA tubes were stored at -80℃ until miRNA extraction. Heparin tubes were immediately centrifuged at 3,000×g for 10 min at 4℃. Extracted plasma was stored at -80℃ until progesterone assay.” 

The rectal temp. was measured with three technical replications when the blood was sampled as described in the method section, but other parameters such as body weight, milk yield, and ambient temperature/humidity were measured and recorded daily using automatic measuring instruments as described in the method section. 

In Fig 1, you may find the date of sampling and information about THI; 15th Aug, and 17th Oct.

C2: As parameter of exposure to heat was chosen THI in barn. Were cows at any time exposed to the outside temperatures?

Re: Yes, the barn was opened to outside environment, but measuring devices were located inside the barn

C3: Pregnancy as a matter of investigation: were all pregnant cows of the same gestational age?

Re: They were not exactly the same age from two and a half months to three months when the blood samples were collected.

Pls see specific info. in the M&M section; All cows had 227±45.5 (mean±standard deviation) average milking days (Individual cow records including age, parity, and calving date in S1 Table).

C4: In the present model of investigation where comparison of blood samples of the same pregnant cows but in different time period (and different gestational age) were made in order to investigate factor-heat stress could be disputable. Namely, differences observed in different time periods could be related to pregnancy state and not heat stress. Differences in progesterone levels observed also, could be related to the stage of pregnancy and not necessary induced by heat stress.

Re: we added in the discussion section as follow

“As described above, the blood samples were collected from the same animals including pregnant and non-pregnant cows, suggesting that the different period or gestational age may affect the profile of gene expression. Thus, we first identified differential expressed genes in both pregnant and non-pregnant cows under HS, reflecting effects of HS and different time periods including different stage of lactation. However, the effects may be canceled out by the common genes both pregnant and non-pregnant cows, and their parities with different stages of lactation (S1 Table). In addition, profiles of putative target genes by DE miRNAs are also associated with metabolism and reproductive system, supporting that HS may induced the DE miRNAs.”

We agree with your comments regarding progesterone, so we deleted some sentence in abstract, but left in the main text because our findings (elevated P4 in pregnant in HS) means pregnant cows are more susceptible to HS. 

“ Particularly, progesterone concentrations known to have maternal warming effects were at similar levels in non-pregnant cows but significantly increased in pregnant cows under heat stress conditions…

with elevated progesterone concentrations” 

C5: Line 105: the authors stated: “Except for two cows, all cows were similar…” Were these differences taken into account considering results analysis?

Re: No, we deleted this sentence. 

C6: Results: Statistical significances were not labeled neither in Table 1 nor in Figures

Re: we labelled in Table 1 and figures

C7: Line 224: it is not stated under what conditions significance was observed

Re: we added “under HS conditions)”

“the rectal temperatures of pregnant cows (40.15℃±0.17) were higher than that of non-pregnant cows (39.36℃±0.1, P < 0.05, Table 1) under HS conditions”

C8: Line 237: please rephrase, it is not clear at what time point significant difference was observed and please explain …”and milk increased after HS”? compared to what? That’s not what Figure shows.

Re: We rephrased to clarify our findings as follow

“Relative average milk yield (AMY) decreased gradually in both pregnant and non-pregnant cows, compared to milk yield of May, particularly a significant reduction was observed in pregnant cows during extremely hot summer (August), and the recovery of AMY was observed in September (Fig 2A), indicating HS more severely may affect milk production of pregnant cows”

Reviewer #2: Comments to Authors:

Major points

C1: The authors do not seem to statistically analyze the large data set of RNA-seq appropriately, as shown in Line 185-186. For the large data set analysis, the authors should do multiple comparison and test the significance of the difference between the treatments by FDR, Bonferroni, or Benjamini criteria to avoid detection of false positives. They just describe ‘P < 0.05’, but do not mention the statistical analysis. This is of critical importance, which further affects not only subsequent screening and PCR of DE miRNAs, but also bioinformatics analyses. Using the incorrectly processed data, they will fail to reach correct appropriate conclusion and interpretation of the data, which will mislead the readers of this manuscript. 

Re: As reviewer pointed out, a proper statistical analysis would be very important for large dataset such as RNAseq comparison. Moreover, multiple comparison (testing) should have done using FDR, Bonferroni etc. However, test significant by FDR and Bonferroni would be too much conservatives for such a small dataset like our samples. Furthermore, test significance would not tell biological significant. Therefore, we set a significant threshold 0.01. 

We also described statistical procedure in the manuscript (Material and Methods) as below, In addition, our qRT-PCR validate our analysis of RNAseq data

“The differentially expressed gene analysis was performed by Limma-voom v3.34.9 R package (Smyth, 2005; Law et al., 2014; Ritchie et al., 2015). Before the analysis, genes which had raw read counts were filtered out by ‘filterByExpr’ function in edgeR R package (Robinson et al., 2010). Furthermore, TMM normalization was performed to normalize each library size using ‘calcNormFactors’ function in edgeR. Next, we used ‘voom’ function of Limma for read counts to transform to logarithmic (base 2) scale prior to linear modeling. Finally, empirical Bayes and moderated t-test were used to detect differentially expressed genes between two groups. The threshold to identify differentially expressed genes was set up to the p-value < 0.01 and the logarithm fold change |logFC| > 2.”

C2: Furthermore, the R-package for methods to visualize DE miRNAs is not provided.

Re: edgeR was used for differential expression (DE) analysis in R language. Pls see our response above

C3:In addition, comparison of the treatments is not described, for example, no animal numbers for each treatment.

Re: we added them in the experimental animal and blood collection (sub-section of material and methods); 

“Pregnant (n=4) and non-pregnant (n=5) cows were used”

And sampling were performed two times (summer and autumn) “when daily minimum THI > 72 and daily maximum THI <72 lasted for more than four weeks (Fig 1)”

Minor points

C4: Line 41: Define miRNA.

Re: We spelled out; microRNAs (miRNAs)

C5: Line 44: It is obvious that circulating miRNAs do not directly regulate gene expression in the cells of the target tissues. How do the authors explain the action of circulating miRNAs on these target cells?

Re: Admittedly, we do not have direct evidence for the gene regulation by miRNA, but our findings of differential miRNA expression as biomarkers may be involved in or associated with the ovarian function under the heat stress. Therefore, we tried to explain possible function of DE miRNA to biological function. In addition, we can find the similar studies (Pregnancy-Ioannidis and Donadeu 2016; Heat stress-Zhung et al., 2014)

C6:Line 67: Holstein Friesian ?

Re: Yes, we corrected to Holstein Friesian 

C7: Line 91: Define miRNA.

Re: Yes, we corrected to microRNA(miRNA)

C8: Line 104-105: the sentence ‘four of them were pregnant …..3.25.’ does not make sense.

Re: we deleted and added (individual cow records in S1 Table)

C9: Line 141: Define (spell out) NHS.

Re: we checked and spell out; 

“under HS (Heat Stress; THI: 86.29) and NHS conditions (Non-Heat Stress; THI: 60.87)” in Rectal temperature in method section. .

C10: Line 163: Show the reason why the authors chose miR-128 for endogenous control.

Re: We refer to Ioannidis and Donadeu’s study “Changes in circulating microRNA levels can be identified as early as day 8 of pregnancy in cattle”; miR-126 was least variable. We cited this paper in the method section.

---

## [Decision Letter · Decision Letter 1]

17 Jul 2020

PONE-D-20-07553R1

Analysis of circulating-microRNA expression in lactating Holstein cows under summer heat stress

PLOS ONE

Dear Dr. Choi,

Thank you for submitting your manuscript to PLOS ONE. After careful consideration, we feel that it has merit but does not fully meet PLOS ONE’s publication criteria as it currently stands. Therefore, we invite you to submit a revised version of the manuscript that addresses the points raised during the review process.

We look forward to receiving your revised manuscript.

Kind regards,

Juan J Loor

Academic Editor

PLOS ONE

Reviewers' comments:

Reviewer's Responses to Questions

**Comments to the Author**

1. If the authors have adequately addressed your comments raised in a previous round of review and you feel that this manuscript is now acceptable for publication, you may indicate that here to bypass the “Comments to the Author” section, enter your conflict of interest statement in the “Confidential to Editor” section, and submit your "Accept" recommendation.

Reviewer #1: (No Response)

Reviewer #2: (No Response)

2. Is the manuscript technically sound, and do the data support the conclusions?

Reviewer #1: (No Response)

Reviewer #2: Partly

3. Has the statistical analysis been performed appropriately and rigorously? 

Reviewer #1: (No Response)

Reviewer #2: No

4. Have the authors made all data underlying the findings in their manuscript fully available?

Reviewer #1: (No Response)

Reviewer #2: (No Response)

5. Is the manuscript presented in an intelligible fashion and written in standard English?

Reviewer #1: Yes

Reviewer #2: Yes

6. Review Comments to the Author

Reviewer #1: Although improved, some methodological issues remain unclear.

Namely, study results showing different progesteron levels in the same cow with months apart could not be attributed to heat stress only, since progesteron levels show oscilations during pregnancy (Stabenfeldt et al, 1970). Therefore, this should be stated along with other observations related heat stress and pregnancy.

In Discussion, line 402: an adjective “reliable“ should be omitted since the present study did not show that. For such conclusion, larger datasets as well as adequate statistical analyses to detect accuracy of miRs as biomarkers are needed.

All in all, limitations of the present study need to be pointed out: small datasets, study design and statistical approach

Reviewer #2: Line 40-42, 382-385: These sentences should be revised or deleted. The authors mention, "we found that differentially targeted progesterone biosynthesis (StAR) and the function of corpus luteum-related genes (CCL11, XCL)", however, they only predicted the targeted genes by use of bioinformatic databases but did not obtain any evidence showing that the miRNAs target and regulate the potential target genes. In addition, as for almost all the circulating miRNAs, it is difficult to determine which tissues release circulating miRNAs and where they are up-taken. I think, therefore, relationships between circulating miRNAs and the target genes in potential target tissues cannot be easily concluded without any data regarding gene expression in the target tissues.

Line 368-410: Regarding the potential target genes, these paragraphs are with too much description only for speculation from the predicted data, and therefore the volume of the description should be much more reduced. If the authors show transcriptomic data, then they could discuss more about the target genes and the physiological adaptation, however, actually this is not the case here. For this reason, the line 409-410 should be deleted because the authors did not show whether the signaling pathways were responsible for several important regulations in lactating cows under HS conditions or not in fact.

7. PLOS authors have the option to publish the peer review history of their article (what does this mean?). If published, this will include your full peer review and any attached files.

Reviewer #1: No

Reviewer #2: No

---

## [Author Response · Author response to Decision Letter 1]

20 Jul 2020

Reviewer #1: 

C1 :Although improved, some methodological issues remain unclear.

Namely, study results showing different progesteron levels in the same cow with months apart could not be attributed to heat stress only, since progesteron levels show oscilations during pregnancy (Stabenfeldt et al, 1970). Therefore, this should be stated along with other observations related heat stress and pregnancy.

Re: We delete materials & methods, results and discussion section about progesterone.

C2: In Discussion, line 402: an adjective “reliable“ should be omitted since the present study did not show that. For such conclusion, larger datasets as well as adequate statistical analyses to detect accuracy of miRs as biomarkers are needed.

Re: We deleted and revised (pls refer to our response to C5) 

“ the selected miRNA could be potentially used to determine response to HS”

C3: All in all, limitations of the present study need to be pointed out: small datasets, study design and statistical approach

Re: we added in conclusion section. 

“Admittedly, the our findings of the present study are limited by the small datasets , their statistical approach, experimental designs that could be attributed to the inherent limitations of the samples (synchronized gestation ages, parities, lactation stages) and heat stress condition (frequent, intense, and duration). Thus, we need to verify our results using model animals or the climate control system (temperature and humidity)”. 

Reviewer #2: 

C4: Line 40-42, 382-385: These sentences should be revised or deleted. The authors mention, "we found that differentially targeted progesterone biosynthesis (StAR) and the function of corpus luteum-related genes (CCL11, XCL)", however, they only predicted the targeted genes by use of bioinformatic databases but did not obtain any evidence showing that the miRNAs target and regulate the potential target genes. In addition, as for almost all the circulating miRNAs, it is difficult to determine which tissues release circulating miRNAs and where they are up-taken. I think, therefore, relationships between circulating miRNAs and the target genes in potential target tissues cannot be easily concluded without any data regarding gene expression in the target tissues.

Re: we deleted line 38-42, and revised. Pls see our response to C5, We mentioned predicted genes/function via bioinformatics analysis

C5: Line 368-410: Regarding the potential target genes, these paragraphs are with too much description only for speculation from the predicted data, and therefore the volume of the description should be much more reduced. If the authors show transcriptomic data, then they could discuss more about the target genes and the physiological adaptation, however, actually this is not the case here. For this reason, the line 409-410 should be deleted because the authors did not show whether the signaling pathways were responsible for several important regulations in lactating cows under HS conditions or not in fact.

Re: We rewrote the following paragraphs to achieve a more concise description. 

Integrative approaches using the predicted target genes of DE miRNAs, and DAVID identified several KEGG pathways such as FoxO signaling, actin cytoskeleton, Rap1 and rapamycin (mTOR) signaling that are involved in transcription activation of heat shock proteins, mitochondria distribution and activity, regulation of energy homeostasis including glucose, and lipid metabolism [49-57]. We also found, in line with a previous study [25], immune response related pathways such as chemokine and TNF signaling [58, 59]. In addition, several DE miRNAs (bta-miR-146b, bta-miR-20b, bta-miR-29d-3p, bta-miR-1246) were differentially expressed in summer heat-stressed pregnant cows. Interestingly, bioinformatics analysis showed their predicted target genes are associated with function of corpus luteum including progesterone biosynthesis [60-61] and prolactin signaling pathways (Table S4). 

Comprehensively, our findings suggest that the experimentally verified miRNA targets and in silico analysis reflect that the selected miRNA could be potentially used to determine response to HS. 

Conclusions

We analyzed physiological HS indicators such as milk yield, rectal temperature, ADG, in pregnant and non-pregnant cows and found that pregnant cows are more vulnerable under HS conditions. In transcriptome analysis of miRNAs using RNA-sequencing, 11 miRNAs (bta-miR-19a, bta-miR-19b, bta-miR-30a-5p, several from bta-miR-2284 family) were differentially expressed (|FC| ≥ 2, P < 0.05) in both pregnant and non-pregnant cows. In pregnant cows under HS, two miRNAs (bta-miR-20b, bta-miR-29d-3p) were upregulated while two miRNAs (Bta-miR-146b, bta-miR-1246) were downregulated. These selected miRNAs could be potential biomarkers associated with HS. Admittedly, the our findings of the present study are limited by the small datasets , their statistical approach, experimental designs that could be attributed to the inherent limitations of the samples (synchronized gestation ages, parities, lactation stages) and heat stress condition (frequent, intense, and duration). Thus, we need to verify our results using model animals or the climate control system (temperature and humidity).

---

## [Decision Letter · Decision Letter 2]

19 Aug 2020

Analysis of circulating-microRNA expression in lactating Holstein cows under summer heat stress

PONE-D-20-07553R2

Dear Dr. Choi,

We’re pleased to inform you that your manuscript has been judged scientifically suitable for publication and will be formally accepted for publication once it meets all outstanding technical requirements.

Kind regards,

Juan J Loor

Academic Editor

PLOS ONE

Additional Editor Comments (optional):

Reviewers' comments:

Reviewer's Responses to Questions

**Comments to the Author**

1. If the authors have adequately addressed your comments raised in a previous round of review and you feel that this manuscript is now acceptable for publication, you may indicate that here to bypass the “Comments to the Author” section, enter your conflict of interest statement in the “Confidential to Editor” section, and submit your "Accept" recommendation.

Reviewer #1: All comments have been addressed

Reviewer #2: All comments have been addressed

2. Is the manuscript technically sound, and do the data support the conclusions?

Reviewer #1: Yes

Reviewer #2: Yes

3. Has the statistical analysis been performed appropriately and rigorously? 

Reviewer #1: (No Response)

Reviewer #2: N/A

4. Have the authors made all data underlying the findings in their manuscript fully available?

Reviewer #1: (No Response)

Reviewer #2: Yes

5. Is the manuscript presented in an intelligible fashion and written in standard English?

Reviewer #1: (No Response)

Reviewer #2: Yes

6. Review Comments to the Author

Reviewer #1: The authors responded adequately to remarks. I have no further suggestions. I suggest the acceptance of the manuscript

Reviewer #2: Although this study would be much improved by an integrative investigation of mRNA-miRNA transcriptomic relationship with data of mRNA-seq or microarray, the authors have appropriately addressed to the comments in this revision.

7. PLOS authors have the option to publish the peer review history of their article (what does this mean?). If published, this will include your full peer review and any attached files.

Reviewer #1: No

Reviewer #2: No

---

## [Editor Report · Acceptance letter]

21 Aug 2020

PONE-D-20-07553R2 

Analysis of circulating-microRNA expression in lactating Holstein cows under summer heat stress 

Dear Dr. Choi:

I'm pleased to inform you that your manuscript has been deemed suitable for publication in PLOS ONE. Congratulations! Your manuscript is now with our production department. 

Kind regards, 

on behalf of

Dr. Juan J Loor 

Academic Editor

PLOS ONE